# Cultural adaptation and validation of the short food literacy questionnaire (SFLQ) for adults in Lebanon

Sahar Obeid[1‡], Souheil Hallit[2,3,4‡], Feten Fekih-Romdhane[5,6], Marie Hokayem[7], Ayoub Saeidi[8], Yonna Sacre[7*], Maha Hoteit[9,10,11*]

1 Department of Psychology and Education, School of Arts and Sciences, Lebanese American University, Byblos, Lebanon, 2 School of Medicine and Medical Sciences, Holy Spirit University of Kaslik, Jounieh, Lebanon, 3 Department of Psychology, College of Humanities, Effat University, Jeddah, Saudi Arabia, 4 Applied Science Research Center, Applied Science Private University, Amman, Jordan, 5 The Tunisian Center of Early Intervention in Psychosis, Department of Psychiatry "Ibn Omrane", Razi hospital, Manouba, Tunisia, 6 Faculty of Medicine, Tunis El Manar University, Tunis, Tunisia, 7 Department of Nutrition and Food Sciences, Faculty of Arts and Sciences, Holy Spirit University of Kaslik (USEK), Jounieh, Lebanon, 8 University of Kurdistan, Department of Physical Education and Sport Sciences, Faculty of Humanities and Social Sciences, Sanandaj, Kurdistan, Iran, 9 PHENOL Research Program, Faculty of Public Health, Section 1, Lebanese University, Beirut, Lebanon, 10 Department of Primary Care and Population Health, University of Nicosia Medical School, Nicosia, Cyprus, 11 INSPECT-LB (Institut National de Santé Publique, d'Épidémiologie Clinique et de Toxicologie-Liban), Beirut, Lebanon

‡ SO and SH are first co-authors on this work.
* maha.hoteit@cnrs.edu.lb (MH); yonnasacre@usek.edu.lb (YS)

## Abstract

### Introduction

In recent decades, Food Literacy (FL) has gained significant attention in the fields of food and nutrition. It is considered a key determinant of public health and presents a promising approach to addressing health challenges like obesity and environmental sustainability. However, Lebanon currently lacks a validated tool to assess food literacy. In this context, the Short Food Literacy Questionnaire (SFLQ) offers potential for translation and cultural adaptation. Adapting an established tool for use in a new cultural setting enables cross-country comparisons and supports the development of cross-cultural study models. Therefore, this study aims to culturally adapt the SFLQ to assess food literacy among Lebanese adults, enhance their knowledge, and promote healthier lifestyle habits.

### Methods

A nationally representative sample of 450 adults was selected through probability cluster sampling from the eight Lebanese governorates. The SFLQ was administered between 11 December 2022 and 23 March 2023 to evaluate participants' food literacy. Confirmatory Factor Analysis was conducted using SPSS AMOS version 30, applying the maximum likelihood estimation method to obtain parameter estimates.

---

**Data availability statement:** All relevant data are within the manuscript and its Supporting Information files available on: Hoteit, M. (2025, August 29). Adults_Food literacy. Retrieved from osf.io/kdpf2.

**Funding:** The author(s) received no specific funding for this work.

**Competing interests:** The authors have declared that no competing interests exist.

## Results

The average age of the participants was 46.0 years, with women making up 59.0% of the sample. The one-factor model showed an acceptable fit after accounting for correlated residuals between items 4−5, 9−10 and 11−12 (RMSEA = 0.079 (90% CI 0.068, 0.091), SRMR = 0.053, CFI = 0.935, TLI = 0.916). Internal consistency was satisfactory ($\omega$ = .86/ $\alpha$ = .86). Measurement invariance across gender was confirmed at the configural, metric and scalar levels. Males had significantly higher average SFLQ scores than females (32.80 ± 7.91 vs 28.76 ± 9.26; $p < 0.001$, Cohen's d = 0.486). A significant negative correlation was observed between SFLQ and Household Food Insecurity Access scores ($r = −0.28$; $p < 0.001$).

## Conclusion

The SFLQ demonstrated strong internal consistency, indicating that it is a reliable tool for both research and clinical use. Expanding validation efforts to include a broader Arabic-speaking population, particularly individuals without internet access, would further strengthen the tool's applicability and cultural relevance across diverse contexts in the Arab world.

## Introduction

### Food literacy

Over the past 25 years, Food Literacy (FL) has emerged as an increasingly important concept within food and nutrition research. Recognized as a crucial factor in promoting public health, FL offers a valuable framework for tackling complex issues such as obesity and environmental sustainability [1]. It encompasses the knowledge and skills needed to make informed dietary decisions, which includes an understanding of food systems and the practical abilities involved in choosing, preparing and consuming food. FL has also been described as a dynamic process that enables individuals, families and communities to safeguard dietary quality and build resilience in the face of changing conditions [2,3].

FL is closely linked to the broader concept of health literacy and follows a similar theoretical model. According to Nutbeam (1998), health literacy involves the cognitive and social skills that influence a person's ability an motivation to access, understand and apply information for maintaining good health" [4] (p.357). Like health literacy, FL goes beyond basic understanding, incorporating the motivation and capability to critically engage with health-related content [5].

Six main dimensions are generally used to define FL: "Skills and Behaviors, Food/Health Choices, Culture, Knowledge, Emotions, and Food Systems" [6].

### Development of food literacy and correlates

Food literacy develops throughout an individual's life, with its transmission often beginning within the family [7,8]. In addition to family influence, schools also play a

crucial role in promoting food literacy, either through integration into the general curriculum or via specific programs and interventions [9].

As previously noted, parents significantly shape their children's dietary preferences and eating behaviors by providing access to specific foods and modeling eating habits through their own actions [10]. Nutritional awareness and the ability to understand food-related information- two essential aspects of food literacy- are closely associated with the overall dietary quality within households [11]. Therefore, enhancing food literacy across the population is essential for encouraging healthier food choices, which in turn can lead to better nutrition and health outcomes [12].

Consequently, a food-literate individual makes well-informed decisions that support both personal health and the sustainability of the food system, taking into account environmental, social, economic, cultural, and political factors [13]. Research by Begley et al. has shown that households facing food insecurity often demonstrate lower levels of food literacy, reflected in low cooking self-confidence and unhealthy food purchasing habits [14]. Studies also showed that FL is positively associated with adherence to dietary guidelines [15,16] and negatively correlated with the consumption of fast foods [17], making it a crucial determinant of healthier diets.

## Short food literacy questionnaire

Various questionnaires are designed to systematically assess different aspects of food literacy, such as the Short Food Literacy Questionnaire (SFLQ) and the Self-Perceived Food Literacy (SPFL) Questionnaire. The SFLQ emphasizes personal skills, such as assessing and interpreting nutritional information, which are crucial for making informed and healthy food decisions [18].

The SFLQ was initially created by Krause et al. in Switzerland as part of a research project aimed at reducing salt intake among workers [18]. It followed a unidimensional structure and demonstrated adequate construct validity and internal reliability. The 12-item questionnaire evaluates various abilities, including locating and interpreting nutrition-related information, understanding content from different sources, familiarity with national dietary recommendations, preparing nutritious meals, supporting others with nutrition concerns, judging the credibility of food information and advertisements, and recognizing the health benefits or long-term effects of specific foods [18].

The SFLQ was later adapted into Turkish [19] and Polish [20], which also confirmed its reliability and validity.

## The present study

This study seeks to culturally adapt the SFLQ for use in Lebanon, to better assess adults' food literacy, enhance their knowledge, and encourage healthier lifestyle choices. Understanding food literacy in Lebanon is particularly relevant given the region's high rates of food insecurity [21] and obesity, and the absence of validated tools for measuring food literacy in Arabic-speaking populations. In the Middle East and North Africa (MENA) region, food literacy has been found to influence various factors, including eating habits, the use of food labels, academic outcomes, nutritional variety, nutrient consumption and food security status [22]. According to the Food and Agriculture Organization (FAO), around 33% of the Arab population experienced food insecurity in 2020 [23]. There were 69 million undernourished individuals in the Arab world by 2020 [23], with estimates that this figure could exceed 75 million by 2030 [23]. At the same time, obesity rates in the region surged to 28.8% in 2020, placing the Arab region as the third most affected by obesity worldwide [23].

Research on nutrition literacy and food literacy in the MENA region remains limited. This gap is highlighted in a recent review (27), which urges regional researchers to begin addressing these issues. In Lebanon, a few preliminary studies have been conducted. One study [24], carried out in 2022 among 450 parent-adolescent pairs, found that approximately 45% of adolescents had low nutritional literacy, while nearly half of the parents (47.8%) demonstrated poor food literacy. Furthermore, the study found that 68.2% of households and 54% of adolescents experienced food insecurity. Among adolescents, 6.7% were stunted, 4.7% were underweight, 32.2% were overweight or obese, and 16.7% were anemic. Another study [25], involving college students from both public and private universities in Lebanon, showed that around half of the participants reported rarely or never

consuming a balanced diet (55.6%) or eating healthy foods (47.7%). Additionally, less than half of the students engaged in food literacy behaviors such as selecting healthy foods (37.5%) or planning meals (33.2%). In contrast, most students often or always ate convenient foods (82.9%) and prepared meals quickly (71.9%). Moreover, a cross-sectional study [26] involving ten Arab countries revealed that Lebanon ranked second in terms of highest rates of nutrition illiteracy (37.4%).

Having said that, these findings are especially relevant in Lebanon, where food insecurity has become widespread, affecting nearly all households [27–29]. Lebanese people are facing significant challenges in recovering from prolonged economic hardships that impact both daily shopping and family meals. Addressing these challenges requires urgent efforts to enhance adults' food literacy and eating habits through targeted education and management strategies [30]. Before introducing specialized nutritional care and educational programs for specific groups, it is important to first understand their perceived nutritional and food literacy. Adding to that, there is currently no validated tool in Lebanon to assess food literacy. In this context, the SFLQ could be translated and adapted for use. Thus, adapting an instrument developed in other countries for use in a new cultural context facilitates cross-country comparisons and the development of cross-cultural study models. We hypothesize that the Short Food Literacy Questionnaire will: (1) confirm the original unidimensional factor structure, (2) show high internal consistency and measurement invariance across genders, and (3) demonstrate meaningful association with food insecurity.

## Methods

### Ethical considerations

The study was carried out in alignment with the ethical principles outlined in the Declaration of Helsinki. Ethical clearance was obtained from the Ethics Committee of Al-Zahraa University Medical Center, Lebanon (Reference # 10-12-2022). All participants were informed about the purpose of the study and provided written consent prior to their involvement. Pparticipation was voluntary, with no penalty upon withdrawal.

### Study design

A cross-sectional study design was employed, targeting a nationally representative sample of adults residing in Lebanon (between 11 December 2022 and 23 March 2023). Participants were selected through a probability-based cluster sampling approach, with recruitment taking place across the eight Lebanese governorates. Fig 1 presents the distribution of the sample across regions. Eligibility criteria required participants to be Lebanese citizens, aged 18 and above and free from any chronic diseases. Individuals with chronic illnesses were excluded because such conditions can significantly influence food choices, eating behaviors, and nutritional requirements, which could influence the assessment of general food literacy. Moreover, from each household, only one adult was recruited after dissemination of the survey announcement in several public areas, social medias and healthcare settings [31].

***The Short Food Literacy Questionnaire***: The validated 12-item SFLQ was used to assess adults' food literacy. The questionnaire was translated into Arabic and reviewed by experts. It is self-administered and uses four- or five-point Likert scales, with total scores ranging from 7 to 52, where higher scores indicate better food literacy. Response options include: "1=strongly disagree" to "4=strongly agree," "1=very poor" to "5=very good," "1=very difficult" to "4=very easy," and "1=never" to "5=always." A median score of 31.0 was used to categorize adults as having poor or adequate food literacy [24]. The scale was forward and backward translated to Arabic by two different translators. The original and translated English versions were then compared by the research team and the two translators to solve any discrepancies. The research team and the translators resolved minor discrepancies. A pilot study was done on 30 adolescents to make sure that all questions are clear to them; no changes were done afterwards.

***Household Food Insecurity Access Scale (HFIAS):*** It is a widely used, validated tool developed by the Food and Nutrition Technical Assistance (FANTA) project to assess the access component of household food insecurity, particularly in low- and middle-income settings. The HFIAS consists of nine occurrence questions that capture households' experiences related to anxiety about food, insufficient quality, and insufficient food intake, each followed by a

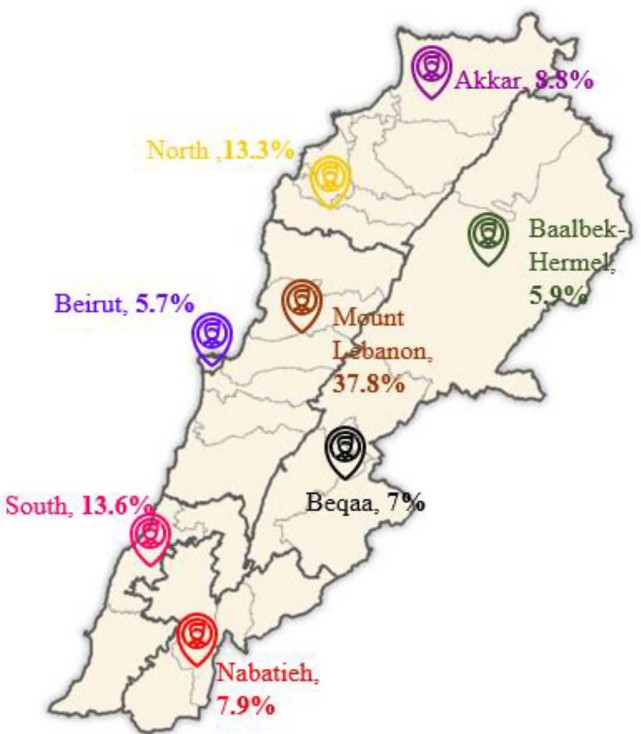

**Fig 1. Distribution of study participants across governorates [32].**

frequency-of-occurrence question if the answer is affirmative. Scoring ranges from 0 to 27, with higher scores indicating more severe food insecurity [33]. Additionally, households can be categorized into four levels: food secure, mildly, moderately, and severely food insecure. This tool is especially valuable for tracking the prevalence and severity of food insecurity and evaluating interventions, although it does not measure nutritional adequacy or individual-level food insecurity. Its simplicity, focus on experiential aspects, and sensitivity to change over time make it suitable for large-scale surveys, such as the one referenced in the study by Nasreddine et al. (2022) [34], which used the HFIAS to examine household food insecurity and its association with dietary patterns and nutritional outcomes in Lebanon.

## Minimum sample size

A Monte-Carlo simulation-based approach described in prior methodological work [35] indicates that, for a one-factor model with 12 items and moderate loadings (0.40–0.70) according to the original validation paper [18], a sample size of approximately 180 is sufficient to achieve stable estimates and adequate power.

## Survey design

The survey was self-administered in Arabic and distributed face-to-face and online. Data collectors were trained public health researchers who conducted in-person recruitment and administration of the questionnaire in public areas and healthcare settings, while the online version was shared via institutional social media platforms and community groups to ensure wide geographical reach. For participants who required clarification during in-person data collection, assistance was provided by the data collectors to ensure comprehension without influencing responses. Only one adult per household was invited to participate. Due to the dual approach (in-person and online), a precise response rate could not be

calculated, especially for online distribution; however, for the in-person recruitment component, the response rate was approximately 78%, with 340 completed questionnaires out of 435 eligible approached adults.

## Statistical analysis

CFA was conducted using SPSS AMOS version 30. The maximum likelihood method was employed to derive parameter estimates. Model fit was assessed using several indices: Root Mean Square Error of Approximation (RMSEA ≤ 0.08), Standardized Root Mean Square Residual (SRMR ≤ 0.05), and both Tucker-Lewis Index (TLI) and Comparative Fit Index (CFI) with acceptable thresholds set at ≥.90 [36]. Convergent validity was examined through the Average Variance Extracted (AVE), with values ≥ .50 considered satisfactory [37]. Due to the initial lack of multivariate normality, a non-parametric bootstrapping approach was applied.

To assess measurement invariance across genders, a multi-group CFA was performed at the configural, metric, and scalar levels [38,39]. Evidence for invariance was supported by ΔCFI ≤ .010 and ΔRMSEA ≤ .015 or ΔSRMR ≤ .010 [40]. Differences in SFLQ scores between males and females were analyzed using the Student's t test.

Internal consistency was evaluated using McDonald's ω and Cronbach's α, with thresholds of ≥ 0.70 indicating satisfactory reliability [41]. Normal distribution of the SFLQ scores was confirmed by skewness and kurtosis values falling within the −1 and +1 range [42]. Relationship between SFLQ scores and Household Food Insecurity Access Scale was assessed using Pearson correlation.

# Results

The mean age of the 442 participants enrolled in this study was 45.07 ± 7.33 years, of whom 27.8% were females. Additional descriptive statistics are detailed in Table 1.

## Confirmatory factor analysis

The fit indices were poor at first (RMSEA = 0.133 (90% CI 0.122, 0.144), SRMR = 0.077, CFI = 0.807, TLI = 0.764) but improved after adding correlations between residuals of items 4–5, 9–10 and 11–12 (RMSEA = 0.079 (90% CI 0.068, 0.091), SRMR = 0.053, CFI = 0.935, TLI = 0.916). The standardised estimates of factor loadings were all adequate (Fig 2). Internal reliability was adequate for the total score (ω = .86/ α = .86).

**Table 1. Sociodemographic and other characteristics of the sample (N = 442).**

| Variable | N (%) |
|---|---|
| Gender | |
| Male | 319 (72.20%) |
| Female | 123 (27.80%) |
| Marital status | |
| Single | 31 (7.00%) |
| Married | 411 (93.00%) |
| Education level | |
| Primary or less | 73 (16.50%) |
| Complementary | 117 (26.50%) |
| Secondary | 111 (25.10%) |
| University | 141 (31.90%) |
| | **Mean ± SD** |
| Age (years) | 45.07 ± 7.33 |
| Household crowding index (person/room) | 1.28 ± 0.78 |

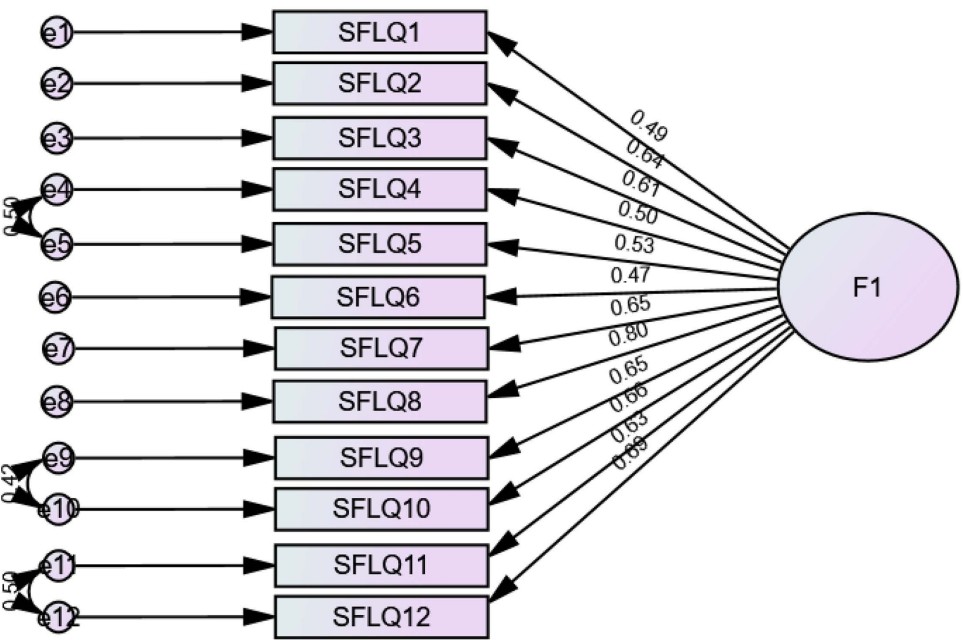

**Fig 2. Standardized loading factors deriving from the confirmatory factor analysis of the Short Form Literacy Questionnaire for adults (parents).**

## Gender invariance

Invariance was shown at the metric and scalar levels in terms of genders (Table 2). A significantly higher mean SFLQ score was found in males compared to females ($32.80 \pm 7.91$ vs $28.76 \pm 9.26$; $p < 0.001$, Cohen's d = 0.486).

## Concurrent validity

The SFLQ score was significantly and negatively associated with the Household Food Insecurity Access score (r = −0.28; $p < 0.001$).

## Discussion

The Arabic validation of the SFLQ represents a significant step forward in evaluating food literacy among Arabic-speaking adults, confirming its reliability and validity and demonstrating its value for research and clinical applications. The study's findings reveal that the 12-item version of the SFLQ, with a one-factor structure, exhibited excellent psychometric properties.

**Table 2. Measurement Invariance of the Short Food Literacy Questionnaire across gender.**

| Model | CFI | RMSEA | SRMR | Model Comparison | ΔCFI | ΔRMSEA | ΔSRMR |
|---|---|---|---|---|---|---|---|
| Configural | 0.925 | 0.061 | 0.061 | | | | |
| Metric | 0.920 | 0.059 | 0.066 | Configural vs metric | 0.005 | 0.002 | 0.005 |
| Scalar | 0.904 | 0.062 | 0.066 | Metric vs scalar | 0.016 | 0.003 | <0.001 |

*Note.* CFI = Comparative fit index; RMSEA = root mean square error of approximation; SRMR = Standardised root mean square residual.

## Factor structure

Initial fit indices (RMSEA = 0.133, CFI = 0.807, TLI = 0.764, SRMR = 0.077) indicated the need for model modification. After adding correlations between the residuals of specific items (4–5, 9–10, and 11–12), all fit indices significantly improved to acceptable levels (RMSEA = 0.079, SRMR = 0.053, CFI = 0.935, TLI = 0.916), indicating a strong model fit.

In line with earlier research, our results support a unidimensional structure of the Arabic adaptation of the SFLQ, in agreement with the original German version of the tool [18], its Turkish [19] and Brazilian [12] validations. However, these findings contrast with previous validations, such as the Polish validation, which revealed a multidimensional structure of the SFLQ in a representative sample of internet users in Poland [20]. Overall, the Arabic SFLQ demonstrates a stable unidimensional structure across languages, supporting cross-national comparisons.

## Internal consistency

Our findings showed strong reliability (ω = .86/ α = .86), comparable to previous studies (Polish α = 0.768 [20], Turkish α = 0.803 [19], Brazilian ω = 0.874 [12], and the original one α = 0.82 [18]). This supports the Arabic SFLQ as a precise and dependable measure of food literacy.

## Gender invariance

In the current study, males scored higher on the SFLQ than females. This contrasts with prior findings suggesting women often show higher food literacy due to their greater involvement in food preparation and healthier eating habits [43,44]. In Turkey [44], a context similar to Lebanon [45], women reportedly have stronger roles in food selection and label use [46,47]. The unexpected male advantage in our sample may reflect cultural, socioeconomic or sampling factors and warrants further investigation. Further research with larger, more diverse samples should explore gender-specific determinants of food literacy to guide targeted education strategies.

## Concurrent validity

Our study revealed a significant negative correlation between SFLQ and Household Food Insecurity Access scores, consistent with previous research [14]. Food literacy plays a crucial role in empowering individuals to provide themselves and others with nourishing meals that support optimal nutrition [13]. Studies have demonstrated a reciprocal relationship between food security and food literacy, where insufficient food literacy can contribute to food insecurity, and food insecurity can impair the ability to practice food literacy behaviours necessary for maintaining a healthy diet [48]. Furthermore, findings have linked food insecurity to poor diet quality, often due to a lack of knowledge about healthier food choices [49]. Diets that lack essential nutrients are key contributors to the development of chronic health issues (obesity, diabetes) [50]. These findings underscore that food insecurity not only leads to malnutrition but also arises from limited awareness or understanding of healthy food choices, ultimately contributing to a range of health issues

## Limitations

The cross-sectional design and reliance on self-reported data limit the strength of the conclusions. The study was conducted solely in Lebanon, which limits the ability to generalize the findings to other Arab-speaking populations. Additionally, potential biases arising from the online survey format and unexamined regional differences within Lebanon may have further impacted the applicability of our findings. Furthermore, the use of snowball sampling for participant recruitment may have introduced selection bias, further limiting the generalizability of the findings. Finally, some psychometric properties, such as test-retest reliability, convergent and divergent validity, as well as content and criterion (predictive) validity, were not tested.

## Clinical implications

Despite certain limitations, the SFLQ is a brief, practical instrument that can inform public health interventions and monitor food literacy in adults. it can be integrated into eHealth or mHealth programs and used to design interventions that build not only knowledge but also practical skills for healthier eating. Finally, this study offers clinicians and researchers a valid and reliable tool for assessing food literacy in Lebanese adults, expanding research opportunities within Arabic-speaking populations.

## Conclusion

This study showed the psychometric properties of the Arabic version of the SFLQ, confirming its reliability and validity as an effective instrument for assessing food literacy in Lebanon. However, the sample was recruited primarily through online platforms, which may have led to an urban bias and an overrepresentation of individuals with higher education levels, thereby limiting the generalizability of the findings. Future research should extend the validation of this tool to a broader and more diverse Arabic-speaking population, including those with limited internet access, in rural settings and with various educational levels. This would further enhance its generalizability and cultural relevance across the Arabic-speaking world.

## Supporting information

**S1 File. National Food Literacy project.**
(XLSX)

## Author contributions

**Conceptualization:** Yonna Sacre, Maha Hoteit.

**Data curation:** Feten Fekih-Romdhane, Marie Hokayem, Yonna Sacre, Maha Hoteit.

**Formal analysis:** Sahar Obeid, Souheil Hallit, Feten Fekih-Romdhane, Marie Hokayem, Ayoub Saeidi, Maha Hoteit.

**Methodology:** Maha Hoteit.

**Project administration:** Maha Hoteit.

**Resources:** Ayoub Saeidi.

**Supervision:** Souheil Hallit.

**Writing – original draft:** Sahar Obeid.

**Writing – review & editing:** Souheil Hallit, Feten Fekih-Romdhane, Marie Hokayem, Ayoub Saeidi, Yonna Sacre, Maha Hoteit.

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
