## [Decision Letter · Decision Letter 0]

24 Jul 2025

Dear Dr. Hoteit,

Thank you for submitting your manuscript to PLOS ONE. After careful consideration, we feel that it has merit but does not fully meet PLOS ONE’s publication criteria as it currently stands. Therefore, we invite you to submit a revised version of the manuscript that addresses the points raised during the review process.

We look forward to receiving your revised manuscript.

Kind regards,

Hasan Durmus

Academic Editor

PLOS ONE

Additional Editor Comments:

Thank you for submitting your manuscript entitled “Cultural Adaptation and Validation of the Short Food Literacy Questionnaire (SFLQ) for Adults in Lebanon” to PLOS ONE. After careful consideration and peer review, I find that your manuscript presents a well-structured and scientifically relevant contribution to the field of public health and nutrition literacy, particularly within Arabic-speaking populations.

However, based on the reviewers’ evaluations and my own assessment, revisions are necessary before the manuscript can be considered for publication.

Gender differences: The finding of significantly higher SFLQ scores among males is notable. This section would benefit from a deeper discussion, including a more robust comparison with existing literature on gender and food literacy.

Limitations section: While the limitations are generally well presented, more emphasis should be placed on the possible bias introduced by using online surveys, especially regarding accessibility and representativeness.

Reviewers' comments:

Reviewer's Responses to Questions

**Comments to the Author**

1. Is the manuscript technically sound, and do the data support the conclusions?

Reviewer #1: Yes

Reviewer #2: Partly

2. Has the statistical analysis been performed appropriately and rigorously?

Reviewer #1: Yes

Reviewer #2: Yes

3. Have the authors made all data underlying the findings in their manuscript fully available?

Reviewer #1: No

Reviewer #2: Yes

4. Is the manuscript presented in an intelligible fashion and written in standard English?

Reviewer #1: Yes

Reviewer #2: No

Reviewer #1: Thank you for your work, it is an important aspect of improving public health. You may consider the following comments:

Overall: The manuscript would benefit from English editing, especially the use of correct grammar and typo mistakes. Some sentences are too long and fragmented, complex to understand.

Introduction: Well-researched. However, it can be more concise. It would be beneficial to create subheadings within this section to enhance readability.

Introduction (2nd to last paragraph): The authors mention “it is important first to understand their

perceived nutritional and food literacy”. If the tool is being standardised to measure FL, what are the reasons behind calling it a perceived literacy?

Methods: The ethical consideration section should move to its correct position as outlined in the author’s instructions

Authors should describe the meaning of all CFA tests and terms used so that readers unfamiliar with the CFA can follow the processes. For example, what is the RMSEA used to determine? And so on

Results: The Authors state in the opening paragraph that about 44% of the participants were women, but the table shows only about 27% of women.

Authors state, “fit indices improved after adding correlations between residuals of items”. Is this an accurate practice in CFA? A further reflection is required on what it means when a good fit is achieved after adding the correlations between the residuals of items. Could this mean that the instrument is not a good fit for the Arab population being studied? This reflection should be added either in the strengths and limitations section or in the discussion section.

Reviewer #2: Thank you for your submission. This is a well-structured and relevant study on the cultural adaptation and validation of the Arabic SFLQ. The methodology is sound, and the psychometric analyses are appropriate. Please clarify the inconsistency in the reported mean age in the Results section and consider expanding the discussion on sample representativeness and limitations. Find the attached comments.

**Do you want your identity to be public for this peer review?** For information about this choice, including consent withdrawal, please see our Privacy Policy

Reviewer #1: **Yes: ** Dr. Paridhi Jha

Reviewer #2: No

---

## [Author Response · Author response to Decision Letter 1]

2 Sep 2025

Comments to authors

This manuscript addresses an important and timely topic: the cultural adaptation and validation of the Short Food Literacy Questionnaire (SFLQ) for use among Arabic-speaking adults in Lebanon.

However, several aspects need improvement:

1. Abstract: The abstract is generally strong but could be improved through minor rewording and slight elaboration in the methods and results section.

We added few ideas to the methods and we edited the abstract for English language.

2. Introduction:

i. The section is too long and text-heavy, Consider breaking long paragraphs and summarizing repetitive points to improve readability.

The introduction was reshuffled as advised by the reviewer. We hope it reads better now.

ii. Some sentences are grammatically awkward or wordy. A light language polish would greatly enhance clarity. Eg There was 69 million undernourished individuals… better to say were instead of was

The paper was edited for English language. Thank you.

iii. Some statements are repeated across paragraphs. Eliminate duplication to tighten the narrative.

The introduction was reshuffled as advised by the reviewer. We hope it reads better now.

iv. The final paragraph begins with a lengthy description of past studies and only introduces this study’s aim toward the very end. Reverse this order: start with the study aim, then explain why this aim is relevant.

Corrected as suggested. Thank you.

3. Method:

i. Criteria: Exclusion of individuals with chronic illness is mentioned but not justified. Why were those with chronic diseases excluded? Clarify.

We added the following explanation to the methods section:

Individuals with chronic illnesses were excluded because such conditions can significantly influence food choices, eating behaviors, and nutritional requirements, which could influence the assessment of general food literacy.

ii. The translation process needs more elaboration:

- Was it forward-backward translated?

- How many experts reviewed it?

- Was there a pretest or pilot? These are essential steps in cultural adaptation and validation.

We apologize for the oversight. We added these ideas to the methods section:

The scale was forward and backward translated to Arabic by two different translators. The original and translated English versions were then compared by the research team and the two translators to solve any discrepancies. The research team and the translators resolved minor discrepancies. A pilot study was done on 30 adolescents to make sure that all questions are clear to them; no changes were done afterwards.

iii. Sample size rule is acceptable, but it would be helpful to mention actual achieved power or use Monte Carlo simulation or RMSEA-based calculations, which are more robust for CFA.

We added the following paragraph to the methods section:

A Monte-Carlo simulation-based approach described in prior methodological work [40] indicates that, for a one-factor model with 12 items and moderate loadings (0.40-0.70) according to the original validation paper [41], a sample size of approximately 180 is sufficient to achieve stable estimates and adequate power.

iv. Data collection is vaguely described (just says public areas, social media, healthcare settings). Who administered the surveys? Online or face-to-face? Self-filled or assisted? Language of administration? Were there response rate details?

Thank you for this valuable comment. The data collection process included both face-to-face and online approaches. Surveys were self-administered in the Arabic language. In public areas and healthcare settings, trained data collectors provided brief instructions and assisted participants when needed, without influencing responses. The online version was distributed through social media platforms using a structured announcement. Due to the combined data collection methods, an exact overall response rate could not be determined; however, for the face-to-face recruitment, the response rate was approximately 78%. We will clarify these details in the revised manuscript.

4. Results: The Results section is generally clear and well-structured. However, there is a serious discrepancy in the reported mean age, which must be corrected. The CFA and measurement invariance analyses are appropriate and well-reported. Minor improvements in table formatting and interpretation of concurrent validity would further enhance the quality and clarity of the results.

eg The mean age is reported as 14.67 ± 2.94 years, which contradicts the descriptive table (Table 1) that reports 45.07 ± 7.33 years. This major inconsistency must be corrected to reflect the accurate demographic.

We apologize for this mistake. It is corrected now.

In Table 1, the percentage for Primary or less is missing a closing parenthesis: 73 (16.5%. Please correct.

Corrected.

Ensure consistency in the presentation of percentages — either include one decimal place throughout or none.

Corrected.

5. Discussion: The Discussion section appropriately interprets the study findings and situates them within existing research. However, it is overly verbose in several areas, with some repetition and dense technical explanation. The authors are encouraged to summarize model fit results more concisely, reconcile unexpected gender differences with prior findings, and expand on the limitations regarding psychometric properties. The clinical and research implications are important and well stated, but would benefit from tighter language. Overall, this section can be significantly improved with focused editing.

We made the changes to the discussion based on the reviewer’s comments. We hope it reads better now.

6. Conclusions

i. Overstatement of "Successful Validation"

Sentence rephrased:

This study showed the psychometric properties of the Arabic version of the SFLQ, confirming its reliability and validity as an effective instrument for assessing food literacy in Lebanon.

ii. Generalization beyond Study Sample: The conclusion appropriately acknowledges the need for broader validation. However, it would be useful to briefly reiterate the limitations of the sample (e.g., online-based, urban bias, higher education level) to ground the claim.

iii. Suggestion for Future Research

Answer to the two previous comments:

We added the following sentences to the conclusion:

CONCLUSION

This study showed the psychometric properties of the Arabic version of the SFLQ, confirming its reliability and validity as an effective instrument for assessing food literacy in Lebanon. However, the sample was recruited primarily through online platforms, which may have led to an urban bias and an overrepresentation of individuals with higher education levels, thereby limiting the generalizability of the findings. Future research should extend the validation of this tool to a broader and more diverse Arabic-speaking population, including those with limited internet access, in rural settings and with various educational levels. This would further enhance its generalizability and cultural relevance across the Arabic-speaking world.

Thank you

Thank you for your time, efforts and comments. We hope the revised version is up to your expectations.

---

## [Decision Letter · Decision Letter 1]

24 Sep 2025

Cultural Adaptation and Validation of the Short Food Literacy Questionnaire (SFLQ) for Adults in Lebanon

PONE-D-25-15423R1

Dear Dr. Hoteit,

We’re pleased to inform you that your manuscript has been judged scientifically suitable for publication and will be formally accepted for publication once it meets all outstanding technical requirements.

Kind regards,

Hasan Durmus

Academic Editor

PLOS ONE

Additional Editor Comments (optional):

Reviewers' comments:

Reviewer's Responses to Questions

**Comments to the Author**

Reviewer #1: All comments have been addressed

Reviewer #2: All comments have been addressed

2. Is the manuscript technically sound, and do the data support the conclusions?

Reviewer #1: Yes

Reviewer #2: Yes

3. Has the statistical analysis been performed appropriately and rigorously?

Reviewer #1: Yes

Reviewer #2: Yes

4. Have the authors made all data underlying the findings in their manuscript fully available?

Reviewer #1: Yes

Reviewer #2: Yes

5. Is the manuscript presented in an intelligible fashion and written in standard English?

Reviewer #1: Yes

Reviewer #2: Yes

Reviewer #1: Thank you for reverting back with this manuscript with suggested modifications. I have no further comments.

Reviewer #2: Thank you for submitting your manuscript to this journal.

All my comments.have been addressed.

Kind regards

Reviewer

**Do you want your identity to be public for this peer review?** For information about this choice, including consent withdrawal, please see our Privacy Policy

Reviewer #1: No

Reviewer #2: **Yes: ** Merga Abdissa Aga

---

## [Editor Report · Acceptance letter]

PONE-D-25-15423R1

PLOS ONE

Dear Dr. Hoteit,

I'm pleased to inform you that your manuscript has been deemed suitable for publication in PLOS ONE. Congratulations! Your manuscript is now being handed over to our production team.

Kind regards,

on behalf of

Dr. Hasan Durmus

Academic Editor

PLOS ONE